# Racial Differences in Perceived Food Swamp and Food Desert Exposure and Disparities in Self-Reported Dietary Habits

**DOI:** 10.3390/ijerph17197143

**Published:** 2020-09-29

**Authors:** Kristen Cooksey Stowers, Qianxia Jiang, Abiodun T. Atoloye, Sean Lucan, Kim Gans

**Affiliations:** 1Rudd Center for Food Policy and Obesity, University of Connecticut, Hartford, CT 06103, USA; abiodun.atoloye@uconn.edu; 2Department of Allied Health Sciences, University of Connecticut, Storrs, CT 06269, USA; 3Department of Human Development and Family Sciences and Institute for Collaboration on Health, Interventions and Policy, University of Connecticut, Storrs, CT 06269, USA; qianxia.jiang@uconn.edu (Q.J.); kim.gans@uconn.edu (K.G.); 4Department of Family and Social Medicine, Albert Einstein College of Medicine, New York, NY 10461, USA; slucan@yahoo.com

**Keywords:** neighborhood environment, food swamps, food deserts, diet quality

## Abstract

Both food swamps and food deserts have been associated with racial, ethnic, and socioeconomic disparities in obesity rates. Little is known about how the distribution of food deserts and food swamps relate to disparities in self-reported dietary habits, and health status, particularly for historically marginalized groups. In a national U.S. sample of 4305 online survey participants (age 18+), multinomial logistic regression analyses were used to assess by race and ethnicity the likelihood of living in a food swamp or food desert area. Predicted probabilities of self-reported dietary habits, health status, and weight status were calculated using the fitted values from ordinal or multinomial logistic regression models adjusted for relevant covariates. Results showed that non-Hispanic, Black participants (*N* = 954) were most likely to report living in a food swamp. In the full and White subsamples (*N* = 2912), the perception of residing in a food swamp/desert was associated with less-healthful self-reported dietary habits overall. For non-Hispanic Blacks, regression results also showed that residents of perceived food swamp areas (OR = 0.66, *p* < 0.01, 95% CI (0.51, 0.86)) had a lower diet quality than those not lving in a food swamp/food desert area. Black communities in particular may be at risk for environment-linked diet-related health inequities. These findings suggest that an individual’s perceptions of food swamp and food desert exposure may be related to diet habits among adults.

## 1. Introduction

Poor diet is one of the top contributors to obesity and many chronic diseases in the United States [1]. Dietary habits may help explain some socioeconomic and racial/ethnic inequities in health [2,3,4]. Though the overall diet quality of US adults has modestly improved compared to the past decades, socioeconomic and racial/ethnic disparities persist [3,5]. For example, a national study found that non-Hispanic Blacks had worse diet quality measured by Healthy Eating Index (HEI) compared to non-Hispanic Whites [3].

Neighborhood food environments play an important role in individuals’ diet quality, weight status, and overall health [6]. “Food desert,” areas where there is little access to healthful food, have long been identified as one possible driver of the obesity epidemic in the US [7,8]. However, recent studies are increasingly shifting their focus on the culpability of “food swamps” (i.e., areas where unhealthy food retailers inundate healthier alternatives) [9]. Previous studies suggest that food swamps are strong predictors of disparities in obesity prevalence among adults [10]. Prior research also points to strong linkages between race and ethnicity, exposure to food swamps and other inequitable food environments, and dietary disparities [10,11,12,13,14,15,16].

Neighborhood food environments are commonly measured by “objective” measures such as geographic information system (GIS)-based measurements and store audits, but accounting for perceptions (e.g., perceived availability and accessibility of food or food stores measures) is arguably more important [17]. Though policies attempting to promote healthy eating have mainly focused on improving geographic proximity to the locations of healthy food outlets, the importance of using self-reported individual perceptions of the neighborhood food environment has been demonstrated [18,19,20]. Perceived accessibility to different food outlets may reflect individuals’ shopping behaviors, dietary behaviors, and nutritional status as well or better than objective measures of proximity [21,22,23,24,25].

Recent studies utilizing self-reported food environment measures found significant associations between healthy food access and diet quality [26,27,28,29,30,31,32,33,34]. However, little is known about how perceived food environment relates to self-reported dietary intake and related health, particularly for racial and ethnic minority groups in the US. This study sought to understand how perceptions of food store availability and accessibility are related to reported diet, weight status, and general health. In particular, the study explores whether living in a perceived food desert and food swamp was more likely to be reported by lower-income or racial and ethnic minority individuals, and if such perceptions are related to lower-quality diets, higher weight, and worse reported health.

## 2. Materials and Methods

The current study is based on an online research sample. Study participants came from Amazon Mechanical Turk (MTurk), a crowdsourcing website that is widely used to obtain high-quality data rapidly and inexpensively [35,36,37,38]. This study utilizes data from 30 survey questions that were distributed on Qualtrics, assessing (1) food store access, (2) dietary habits, (3) perceived weight status, (4) perceived health status, and (5) demographics. The survey also included the following red herring question to identify whether or not participants fully read and engaged in the survey: “How closely do you read survey questions? To show that you’re paying attention, please click the ‘Not closely at all’ option as your answer”. The survey was in English only.

Prior to launching the survey, cognitive interviews were conducted with 10 volunteers to test all survey questions with an emphasis on pretesting the food store access and dietary habits questions. The think-aloud cognitive interview approach [39] was utilized to ensure that individuals were interpreting the questions as intended and that survey questions were measuring the constructs as intended (i.e., content validity). Each interview lasted 45–60 min. Question-wording and overall flow were modified based on the cognitive interviews.

The original national sample of U.S. adults consisted of 6357 participants. Drawing from previous research on enhancing data quality in online survey research by screening for inattentive respondents and “speedy” completion times, participants were excluded if they wrongly answered “red herring” questions [40,41,42] (609 participants excluded). Relatedly, participants were also excluded if their response time was less than 6.5 min (1443 participants excluded based on the quartile), indicating that they did not answer the questions carefully [41,42]. The final sample consisted of 4305 participants. In line with other studies paying “MTurkers” 0.50 to 1.00 USD to complete a survey task, [43,44,45,46] we paid 0.70 USD to each MTurk online survey participant. The Institutional Review Boards of Duke University approved all study procedures and materials. Respondents signed an online consent form to indicate that they agreed to participate in the study.

### 2.1. Measures

#### 2.1.1. Outcome Variables

We have three outcome measures including self-reported dietary quality, weight status, and health status. We used the questionnaire from the Information Resources, Inc. (IRI, Chicago, IL, USA) MedProfiler Health and Wellness Survey which is completed by a nationally representative sample of Nielsen panelists on an annual basis and collects data on topics such as eating habits, health opinions, health concerns, and medical conditions [47]. The survey measures participants’ diet quality by asking them how frequently they eat dairy, five servings of fruits/vegetables each day, dessert or indulgent snacks, whole grains, sugar-sweetened beverages, organic foods, at a fast-food restaurant, at a full-service/sit-down restaurant, and how often they cook at home. The frequency that participants engage in each dietary habit was measured on a six-point scale ranging from never, rarely, sometimes, often, most of the time, and always. Combining these individual categories of dietary habits, we created a diet quality composite score ranging from −12 to 27 that deducted 1–6 points for unhealthy dietary habits (i.e., eating dessert, eating fast food, and drinking sugar-sweetened beverages), and added 1–6 points for the remaining variables. Finally, we collapsed categories based on the distribution of diet quality composite score to create a high (12 to 27, 31.0%), medium (7 to 11, 36.0%), and a low category (−12 to 6, 31.3%). Self-reported weight status was measured in a question asking participants how they described their weight by choosing from the following responses: underweight, about right, slightly overweight, and very overweight. To assess self-reported health quality, we used a question asking participants to rate their health quality as one of five different levels including poor, fair, good, very good, and excellent.

#### 2.1.2. Sociodemographic Characteristics

Demographic characteristics were measured in the survey and included gender; annual household income (lower income (<50k USD) vs. higher income (>50k USD)); race/ethnicity (non-Hispanic White, non-Hispanic Black, non-Hispanic Asian, non-Hispanic other, and Hispanic); education level (high school or less, associate’s degree and some college, bachelor’s degree or higher); current family structure (single without children, single with children, married without children, married with children, life partner without children, and life partner with children); geographic area (living in rural, suburban, or urban area); car ownership (own a car or someone in my house owns a car, do not own a car); region of the United States (Midwest, Northeast, Southeast, Southwest, and West); and age.

#### 2.1.3. Perceived Food Swamp and Food Desert Exposure

To measure perceived food swamp exposure, participants were asked about the number of different food stores they have within one mile from their home (for those living in urban or suburban areas) or five miles (if residing in a rural area) [48]. Food stores included grocery stores, supercenter/club stores, farmer’s markets, full-service/sit-down restaurants, convenience/corner stores, fast-food/limited service establishments, and gas stations with foods. We then created a modified Retail Food Environment Index (mRFEI) score to measure self-reported food swamp exposure based on the following equation: mRFEI = ((grocery store + farmer’s market + full service/sit-down restaurant)/total stores). If the mRFEI score = 0, it was designated as a perceived food desert area; if 0 < the mRFEI score < the median (0.368), it was designated as a perceived food swamp area; otherwise, it was defined a nonfood swamp and food desert area [11,49].

### 2.2. Statistical Approach

Multinomial logistic regression models were employed to assess the likelihood of living in a perceived food swamp area or food desert area (measured with mRFEI). The reference group included individuals not living in a food swamp/desert area. To assess racial and ethnic differences relative to non-Hispanic Whites and between each racial and ethnic minority group, separate multinomial regression models were run, including the following reference groups: non-Hispanic White, non-Hispanic Black, non-Hispanic Asian, non-Hispanic Other, and Hispanic. The estimates from the multinomial regression tests were then used to compute racial and ethnic differences in the relative risk (i.e., probability) of living in a food swamp/desert area.

To examine the relationship between the food environment and the three main outcome variables in both the full sample and three subsamples (i.e., non-Hispanic White, non-Hispanic Black, and Hispanic), we first checked to see if the ordinal logistic regression parallel line assumption held. The ordinal logistic regression model was then chosen to assess the predicted probabilities of perceived diet quality and health quality. However, since the assumption of parallel trend test was violated for perceived weight status, multinomial logistic regression models were chosen for this particular outcome variable. For all regression models, the independent variable is perceived food swamp and food desert exposure. Covariates included family income, race/ethnicity, education level, current family structure, geographic area, car ownership, gender, region of the country, and age. For all three main outcome variables, analyses were also stratified by race and ethnicity. Further, for models including the full sample of respondents, we included race and income interaction terms. We ran all statistical analyses using SPSS version 24 [50].

## 3. Results

The final sample consisted of 4305 participants with 38.0% male, 67.6% non-Hispanic White, 22.2% non-Hispanic Black, 2.0% non-Hispanic Asian, 3.8% Hispanic, 33.6% single, and 30.8% low income. Participants were on average 41.3 years old and about half (48.8%) of them lived in a suburban area. Based on mRFEI measures, 40.7% of study participants reported a retail environment consistent with living in a food swamp, and 6.5% reported a retail environment consistent with living in a food desert. More than half of study participants (58.4%) owned a car or someone in the house owned a car. 41.3% of respondents had a bachelor’s degree or more. See Table 1.

Regarding outcome variables, 72.6% of respondents stated that they had good health quality or above. More than half (55.8%) perceived themselves as slightly or very overweight. For diet quality, we calculated that 31.0% had low diet quality, 36.0% medium diet quality, and 31.3% high diet quality (see Table 1).

Refer to Table 2 for the likelihood that participants perceived living in a food swamp or food desert (measured using mRFEI) by race and ethnicity. The results of multinomial logistic regression models showed that compared to non-Hispanic Whites, non-Hispanic Blacks were 38% more likely to perceive living in a food swamp (RR = 1.38, *p* < 0.001); non-Hispanic Asians were less likely to perceive living in a food swamp than Asians (RR = 0.70, *p* < 0.05). Compared to non-Hispanic Blacks, non-Hispanic Asians were less likely to perceive living in both food swamps (RR = 0.53, *p* < 0.001) and food deserts (RR = 0.13, *p* < 0.05); and Hispanics were less likely to perceive living in a food swamp (RR = 0.8, *p* < 0.05). Compared to non-Hispanic Asians, non-Hispanic Others were more likely to perceive living in both food swamps (RR = 1.59, *p* < 0.05) and food deserts (RR = 7.52, *p* < 0.05). Last, Hispanics were more likely to perceive living in a food swamp than non-Hispanic Others (RR = 1.50, *p* < 0.05). As a robustness check, we also ran multinomial logistic regression models of the likelihood of residing in a food swamp/desert by including food swamp measures based on the average (vs. median) mRFEI and found similar results. The only differences were that the likelihood of residing in a food swamp for non-Hispanic Asian vs. White became statistically insignificant, plus the following likelihood of food desert status results became statistically insignificant: non-Hispanic Asian vs. Black and non-Hispanic Asian vs. non-Hispanic Other.

See Table 3 for the total sample, ordinal logistic regression results showed that residents of food deserts (OR = 0.74, *p* < 0.05, 95% CI (0.58, 0.94)) and food swamps (OR = 0.75, *p* < 0.001, 95% CI (0.66, 0.84)) were significantly more likely to have a lower diet quality score than those not living in a food swamp/food desert area. For non-Hispanic Blacks (*N* = 954), regression results showed that residents of food swamp areas (OR = 0.66, *p* < 0.01, 95% CI (0.51, 0.86)) had a lower diet quality than those of Black Americans not living in a food swamp/food desert. There were no statistically significant differences in the relationship between residing in a food swamp or food desert and diet quality in the Hispanic subsample, or perceived health quality and weight status by food swamp/desert residential status in the total sample or any subsample. Therefore, we did not include the results in our paper. We included the regression models with full covariates in Appendix A.

## 4. Discussion

Our study is among the first to explore the role of perceived food swamp and food desert exposure on self-reported dietary habits, health quality, and weight status in a racially and ethnically diverse, national sample of US adults. Overall, participants living in perceived food swamps and food deserts were more likely to report lower diet quality, but this did not translate to statistically significant results for self-reported weight or health status. Further, we found racial and ethnic disparities in the likelihood of individuals residing in a perceived food swamp or food desert. In addition, results from the current study reveal that the relationship between perceived food swamp or food desert exposure and dietary quality vary among non-Hispanic Whites, non-Hispanic Blacks, and Hispanic populations.

### 4.1. Racial and Ethnic Disparities in Perceived Food Swamp/Desert Status

According to the results of multinomial regression models stratified by race and ethnicity, non-Hispanic Black Americans were more likely to report living in a food swamp than adults identifying as non-Hispanic White, non-Hispanic Asian, or Hispanic. Though few studies have assessed the likelihood of residing in a food swamp by race and ethnicity to date [51], national and local studies across the US found that residents of minority neighborhoods are more likely to categorize their neighborhood as a food desert (i.e., poor access to supermarkets, chain grocery stores, and healthful food products) [24,52,53,54,55]. For example, previous research has revealed that residents living in predominantly Black and Hispanic neighborhoods tend to have lower access to chain supermarkets or large grocery stores, and higher access to fast-food outlets when compared to non-Hispanic, White counterparts [52]. Prior studies also suggest that these inequities in the built, food retail environment can be partially attributed to racial residential segregation [56,57]. The current study builds upon this research by going beyond counts of grocery stores and fast-food establishments to utilize a relative measure of the different types of stores (i.e., mRFEI) within a defined boundary. The findings of this study align with previous research indicating that predominantly Black neighborhoods are less likely to have equal access to healthy and unhealthy food retailers [58]. Another study found that more convenience stores were located near secondary schools in predominantly racial/ethnic-minority neighborhoods compared to predominantly White census tracts [59]. Building upon previous studies suggesting that Black and Hispanics are more likely to live in food deserts [13,56], we found that Black Americans were also more likely to perceive living in a food desert than Asians. Notably, relative to earlier studies, we used perceived food desert measures instead of objective measures of accessibility to food outlets. Still, our findings warrant future efforts to address racial and ethnic disparities in the risk of living in a food desert or food swamp.

### 4.2. Perceived Food Swamp/Desert Status and Diet

Consistent with previous studies using objective neighborhood environment measures [10,60], we found that residents of perceived food deserts and swamps had poorer diets relative to those not living in a food swamp/food desert areas in the full sample. This finding suggests that an individual’s perceptions of food swamp and food desert exposure may be an important correlate of diet quality. However, there is limited evidence supporting the elimination of food deserts as a strategy to address poor diet quality [60]. A recent study found that opening a supermarket in a food desert did not result in people buying healthier food [61]. Recent studies of neighborhood food environment are shifting their focus to the culpability of food swamps [9,51], which may be a stronger predictor of inequities in diet and obesity than food deserts [10,11].

### 4.3. Racial and Ethnic Differences in the Relationship between Perceived Food Swamp/Desert Status and Diet

Though to the authors’ knowledge, few studies have assessed the relationship between living in a food swamp or desert area [62] and diet quality across race/ethnicity groups, studies have indicated that residents of predominantly racial and ethnic minority neighborhoods are most likely to be affected by poor access to healthful food and tend to have lower diet quality [52,53,54,55]. In the Black subsample in our study, food swamp residents reported having a lower diet quality than those not living in a food swamp/food desert area. This finding aligns with previous studies that have consistently shown that neighborhoods with higher proportions of Black residents have fewer supermarkets and more convenience stores and fast-food restaurants [52,56,63]. However, in the Hispanic subsample, we did not find any statistically significant differences in the relationship between food swamps or food deserts and dietary health, weight status, or health outcomes.

There were no statistically significant differences in self-reported health quality and weight status by food swamp residential status in our study. Though there is limited research assessing the relationship between the food environment and perceived health, evidence has suggested diet quality has a strong impact on health outcomes [1,64,65,66,67]. Previous studies assessing the association between the neighborhood environment and weight status were inconsistent [6,17,67,68,69,70,71,72,73,74,75,76,77,78,79,80]. Similar to our study, several studies did not find a statistically significant relationship between neighborhood food environment and obesity rates [71,72,73,74]. However, other studies have shown that neighborhood food environment factors, such as food desert and food swamp status, were significantly associated with individuals’ measured weight status [10,11]. For example, Chen et al. [7] used MedProfiler questions about dietary behaviors (e.g., frequency of consuming high-fiber, high-protein, low-carbohydrate, low-fat, low-salt, and low-sugar diets) and found, even after controlling for home food environment factors, food desert status was associated with obesity [67]. The different findings may be due to error inherent in self-reported measures of weight status and diet quality, and the varied ways that researchers used in measuring the built, food environment [67].

### 4.4. Limitations

The current analysis uses self-reported measures of food store exposure, which may be subject to various measurement errors, such as recall bias or socially desirable responses. The present study also uses a self-reported measure of dietary habits. Although we utilized the previously used and validated IRI MedProfiler measure of diet, the current study would have been strengthened by the ability to collect observational data or more comprehensive self-reports (i.e., 24-h dietary recall or Food Frequency Questionnaire) [68]. In addition, the relatively small sample size of Hispanic participants and the absence of acculturation and country of origin limit our understanding of Hispanic ethnicity as a moderator of the relationship between food swamps and diet, and may explain why we did not find statistically significant results in this subsample. Future surveys should be in Spanish as well as English to increase participation among individuals that identify as Hispanic. Further, the current study was only able to examine correlational relationships as the underlying dataset is cross-sectional. Future work should pursue longitudinal designs and natural experiments to explore causality.

## 5. Conclusions

The racial differences in food environment exposure and its relationship with dietary quality identified in this study are consistent with previous evidence pointing to the existence and impacts of racial residential segregation. It also highlights the importance of policies, planning, and development, plus other structural changes that will provide equal opportunity for improved diet quality among both White and racial and ethnic minority populations. Furthermore, the findings from our study suggest that an individual’s perceptions of food swamp and food desert exposure may be associated with dietary habits among adults in the United States. Although research is showing a decline in racial segregation [69], more policy efforts that address the disparities in access to unhealthy versus healthy food retail outlets across neighborhoods are needed to limit the negative impacts on nutrition and health-related outcomes.

Future research should incorporate how well objective and subjective measures of food swamp exposure align, and which are stronger predictors of inequities in health behaviors or outcomes. Future research should also engage communities to identify objective environmental conditions for intervention targets. Further, additional research is needed to identify best practices for maximizing resident engagement and inclusivity in the context of municipal planning and policy development efforts. The application of these best practices may increase the likelihood that future efforts to develop interventions targeting objective conditions in the local food environment will account for and integrate the perceptions and experiences of individuals that live there. If people perceive the remediation of food desert/swamp conditions, they might experience actual improvement in their weight and health. Still, additional research is needed to understand the effects of racial and ethnic disparities in neighborhood food swamp exposure as it will provide relevant information for decision/policymakers to make informed policy changes.

## Figures and Tables

**Table 1 ijerph-17-07143-t001:** Participant demographics (*N* = 4305).

Variables	Mean (SD) or *N* (%)
Sociodemographic Variables	
Age	41.3 (14.3)
Gender	
Male	1634 (38%)
Female	2666 (61.9%)
Household income	
Lower (annual household income <50k)	2109 (49.0%)
Higher	2177 (50.7%)
Education	
High school or less	861 (20.0%)
Associate’s degree or some college	1657 (38.5%)
Bachelor’s degree or higher	1777 (41.3%)
Race/Ethnicity	
Non-Hispanic White	2912 (67.6%)
Non-Hispanic Black	954 (22.2%)
Non-Hispanic Asian	84 (2.0%)
Non-Hispanic other	173 (4.0%)
Hispanic ^1^	162 (3.8%)
Current family structure	
Single without children	1050 (24.4%)
Single with children	391 (9.1%)
Married without children	309 (7.2%)
Married with children	865 (20.1%)
Life partner without children	169 (3.9%)
Life partner with children	100 (2.3%)
Vehicle ownership	
Own a car or someone in my house owns a car	2513 (58.4%)
Urban/suburban/rural area	
Urban	1239 (28.8%)
Suburban	2102 (48.8%)
Rural	964 (22.4%)
Region	
Midwest	969 (22.5%)
Northeast	855 (19.9%)
Southeast	1313 (30.5%)
Southwest	409 (9.5%)
West	758 (17.6%)
**Neighborhood Food Environment**	
Food desert/swamp area ^2^	
Living in a food desert area Living in a food swamp area	279 (6.5%) 1751 (40.7%)
Not living in a food desert/swamp area	2039 (47.4%)
Outcome Variables	
Diet quality ^3^	
Low	1335 (31.0%)
Medium	1549 (36.0%)
High	1347 (31.3%)
Perceived Health quality	
Poor	202 (4.7%)
Fair	974 (22.6%)
Good	1695 (39.4%)
Very good	1029 (23.9%)
Excellent	401 (9.3%)
Perceived Weight status	
Slightly underweight	317 (7.4%)
About right	1582 (36.7%)
Slightly overweight	1813 (42.1%)
Very overweight	591 (13.7%)

^1^ Hispanic White: 114 (2.7%); Hispanic Black: 12 (0.2%); Hispanic Asian: 4 (0.1%); Hispanic other: 29 (0.7%). ^2^ If 0 < the mRFEI score < the median (0.368), it was designated as a perceived food swamp area; if the mRFEI score = 0, it was designated as a perceived food desert area; otherwise, it was defined not living in a food swamp/food desert area. ^3^ Diet quality was measured on a 6-point scale. We collapsed categories based on the distribution of diet quality score to create a high, medium, and low category.

**Table 2 ijerph-17-07143-t002:** Summary of multinomial logistic regression models predicting the likelihood of living in a perceived food swamp or food desert (measured using Modified Retail Food Environment Index (mRFEI)), by race and ethnicity.

Race/Ethnicity		Food Swamp			Food Desert			
Predictors	OR	95% CI		RR ^5^	OR	95% CI		RR ^5^
		lower	higher			lower	higher	
Non-Hispanic Black ^1^	1.71 ***	1.46	2.00	1.38 ***	1.13	0.82	1.54	1.11
Non-Hispanic Asian ^1^	0.57 *	0.35	0.92	0.70 *	0.13 *	0.02	0.94	0.14
Non-Hispanic Other ^1^	1.20	0.87	1.67	1.11	1.10	0.59	2.06	1.09
Hispanic ^1^	1.09	0.78	1.52	1.05	0.64	0.29	1.41	0.67
Non-Hispanic Asian ^2^	0.33 ***	0.20	0.54	0.53 ***	0.12 *	0.02	0.85	0.13 *
Non-Hispanic Other ^2^	0.70 *	0.50	0.99	0.85	0.98	0.50	1.91	0.98
Hispanic ^2^	0.64 *	0.45	−0.91	0.80 *	0.57	0.25	1.30	0.61
Non-Hispanic Other^3^	2.13 *	1.20	3.78	1.59 *	8.51 *	1.08	67.31	7.52 *
Hispanic ^3^	1.92 *	1.08	3.43	1.50 *	4.97	0.59	41.47	4.64
Hispanic ^4^	0.52 *	0.29	0.93	0.94	0.20	0.02	1.68	0.62

^1^ Relative risk with non-Hispanic White as the reference group. ^2^ Relative risk with non-Hispanic Black as the reference group. ^3^ Relative risk with non-Hispanic Asian as the reference group. ^4^ Relative risk with non-Hispanic Other as the reference group. ^5^ We compared the risks of living in a food swamp area or food desert area among different race groups by computing relative risk ratios. Relative Risk is calculated by dividing the probability of an event occurring for group 1 divided by the probability of an event occurring for group 2. Abbreviations: OR, odds ratios; CI, confidence interval; RR, relative risk ratios; * *p* < 0.05, ** *p* < 0.01, *** *p* < 0.001.

**Table 3 ijerph-17-07143-t003:** Summary of logistic regression models predicting diet quality by residing in a food swamp or food desert (measured using Modified Retail Food Environment Index (mRFEI)), stratified by race.

Independent Variables/Covariates	All			Non-Hispanic White (*N* = 2912)			Non-Hispanic Black (*N* = 954)			Hispanic (*N* = 162)		
	OR	95% CI		OR	95% CI		OR	95% CI		OR	95% CI	
		Lower	Upper		Lower	Upper		Lower	Upper		Lower	Upper
Residing in food swamp ^1^	0.75 ***	0.66	0.84	0.75 ***	0.64	0.87	0.66 **	0.51	0.86	1.53	0.79	2.96
Residing in food desert	0.74 *	0.58	0.94	0.75 *	0.56	0.99	0.81	0.48	1.39	0.25	0.05	1.27
Lower income (vs. higher income)	0.86 *	0.74	0.99	0.85 *	0.73	0.99	0.90	0.68	1.18	1.51	0.78	2.93
Male	0.77 ***	0.69	0.87	0.76 **	0.66	0.88	0.83	0.63	1.09	0.54	0.28	1.06
Age	1.01 ***	1.01	1.01	1.01 *	1.00	1.01	1.02 ***	1.01	1.03	1.02	0.99	1.05
Single without children ^2^	0.57 ***	0.42	0.78	0.58 **	0.40	0.82	0.61	0.27	1.38	0.22	0.04	1.24
Single with children	0.89	0.60	1.32	0.95	0.56	1.61	0.84	0.43	1.65	0.68	0.08	6.02
Married with children	0.81	0.53	1.23	0.84	0.47	1.48	0.77	0.38	1.58	1.11	0.09	14.09
Life partner without children	1.28	0.82	1.98	1.3	0.75	2.30	1.28	0.53	3.14	2.45	0.23	25.81
Life partner with children	1.11	0.74	1.67	1.19	0.70	2.02	0.92	0.45	1.87	0.40	0.04	4.25

^1^ Reference group is not living in a food swamps area; ^2^ Reference group is married without children. Abbreviations: OR, odds ratios; CI, confidence interval; * *p* < 0.05, ** *p* < 0.01, *** *p* < 0.001.

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
