# Peer review of "Racial Differences in Perceived Food Swamp and Food Desert Exposure and Disparities in Self-Reported Dietary Habits"

_ijerph, 2020, doi:10.3390/ijerph17197143_

Round 1

Reviewer 1 Report

Thank you for the opportunity to review the manuscript entitled “Racial differences in perceived food swamp and food desert exposure & disparities in self-reported dietary habits” for the “International Journal of Environmental Research and Public Health”. Using self-reported data from a US MTurk online survey, the authors investigate the external validity of a modified Retail Food Environment Index (mRFEI) for diet quality and stratify by ethnicity. While the article might be of general interest for the readers of International Journal of Environmental Research and Public Health, I would like the following issues to see addressed before publication:

  • ll. 77: Please explain the sampling process in more detail. How were the MTurk workers recruited? In which way might the sample be regarded as nationally representative?
  • ll. 97-99: How does the seemingly arbitrary categorization of the diet quality score affect the regression results? Please investigate. Related to this: Why categorize this metric variable after all? The inclusion of a metric score avoids potential methodological artefacts of categorization.
  • l. 106: Please explain, why USD 25’000 is used as the cut-off between low and middle to high income. Consider using a metric variable instead.
  • ll. 114-124: Please explain, why the median is used as maximum value for the food swamp category, and discuss potential alternatives fitting the definition of a food swamp in ll. 41-42.
  • Table 3: Ordinal logistic regression: Does the essential Parallel Regression Assumption hold? If not, use a multinomial regression model instead or even better, run OLS on the metric diet quality score (see point 2).
  • Analyses on weight status are missing completely. However, for further investigation into the external validity of mRFEI, regression of weight status on mRFEI would be worthwhile.
  • ll. 247-257: A further limitation of the study is that it is only able to identify correlations not causal effects, since the underlying data is cross-sectional.

Author Response

September 9, 2020

Dear Reviewer,

Thank you for allowing us the opportunity to submit a revised draft of our manuscript titled “Racial Differences in Perceived Food Swamp and Food Desert Exposure & Disparities in Self-Reported Dietary Habits” to the International Journal of Environmental Research and Public Health special issue devoted to Food Systems, Food Environment, Diet and Nutrition Related Diseases.

We appreciate the time that you and the reviewers have dedicated to reviewing the manuscript and providing edits. We have made revisions based on each of your suggestions in this revised version of the manuscript. Our responses are detailed below each of your initial comments.

  1. 77: Please explain the sampling process in more detail. How were the MTurk workers recruited? In which way might the sample be regarded as nationally representative?

Thank you for this excellent question. This is a national sample of US adults. It is representative of the broader US adult population in terms of gender, educational attainment, income, and U.S. region. We have revised the text to describe it as a “national sample of U.S. adults” in the abstract and in lines 79 and 197 of the manuscript.

Mturkers were recruited via a job posting (comparable to a recruitment flyer) which described the survey (e.g., topic, completion time, incentive, etc.). Participants were able to opt-in to take the online survey by clicking on the survey job posting listing. We have revised the manuscript to add a more detailed description of the sampling process.

Lines 65-67 and 84-86 now read as follows; “The current study is based on an online research sample. Study participants came from Amazon Mechanical Turk (MTurk), a crowdsourcing website which is widely used to obtain high-quality data rapidly and inexpensively.34–36

In line with other studies using MTurk paying each participant $0.50 to $1.00, 42–45 we pay $0.70 to each MTurk online participant to complete the survey task.

  1. 97-99: How does the seemingly arbitrary categorization of the diet quality score affect the regression results? Please investigate. Related to this: Why categorize this metric variable after all? The inclusion of a metric score avoids potential methodological artefacts of categorization.

Thanks for pointing out this very important issue. We agree with the review that the way that we initially categorized the variable was not appropriate. We revised our approach to draw from the distribution of the diet quality composite score to create a diet quality variable with the following 3 different categories: low (31.0%); medium (36.0%); high (31.3%). To explore how this impacts our regression results, we ran both OLS models (using the metric diet quality score) and ordinal logistic regression analysis (using the categorical version). Our comparison revealed that the results didn’t change.

See lines 102-105 for the changes made.

Finally, we collapsed categories based on the range distribution of diet quality composite score to create a high (124 to 27, 31.0%), medium (71 to 11, 36.0%3), and a low category (-12 to 6, 31.3%0) for diet quality which we used in the analysis.  

  1. 106: Please explain, why USD 25’000 is used as the cut-off between low and middle to high income. Consider using a metric variable instead.

We thank the reviewer for this great question. The original survey question asked participants’ household income in seven categories. Thus, we are unable to convert income to a metric variable. We initially used USD 25’000 (i.e., <25k) as low income to align with the US census. However, based on the reviewer comment, we have recategorized them as two categories based on variable distribution: 1. Lower-income (<50k, 49%); 2. Higher income (>50k, 50.7%). We kept a two-category income variable to simplify the Race X Income interaction terms included in full sample regression models.

The socio-demographic characteristics section (lines 110-119) now reads as follows:

Demographic characteristics were measured in the survey and included: gender; annual household income (lower-income (<$50k) vs. higher-income (>$50k); race/ethnicity (non-Hispanic White, non-Hispanic Black, non-Hispanic Asian, non-Hispanic other, and Hispanic); education level (high school or less, Associates degree and some college, Bachelor’s degree or higher); current family structure (single without children, single with children, married without children, married with children, life partner without children, and life partner with children); geographic area (living in rural, suburban, or urban area); car ownership (own a car or someone in my house own a car, do not own a car); region of the United States (Midwest, Northeast, Southeast, Southwest, and West); and age.

  1. 114-124: Please explain, why the median is used as maximum value for the food swamp category, and discuss potential alternatives fitting the definition of a food swamp in ll. 41-42.

Thank you for this important question. We can see how our original wording was unclear.  The median mRFEI value was chosen as the cut-off for food swamp as this food environment measure assesses the % of healthy food outlets. Lower mRFEI indicates the abundance of unhealthy food outlets which aligns with the definition of food swamp. The median was used as a cutoff point instead of a max value for creating a three-category food swamp variable including food desert, food swamp, and non-food swamp/desert areas.

To address the reviewer’s comments about discussing an alternative, we also created a food swamp variable based on the average mRFEI value as a cutoff point and found similar results in terms of the likelihood of residing in a food swamp/desert and relationship with diet quality.

We have revised the manuscript to clarify both points. See lines 125-30 describing the process used to classify food swamp categories and lines 176-178 (copied below).

Lines 125-130:

We then created a modified Retail Food Environment Index (mRFEI) score to measure self-reported food swamp exposure based on the following equation: mRFEI = [(grocery store+farmer's market+full service/sit-down restaurant)/total stores]. If the mRFEI score = 0, it was designated as a perceived food desert area; if 0 < the mRFEI score < the median (0.368), it was designated as a perceived food swamp area; else, it was categorized as a non-food swamp and food desert area.11,48

Lines 176-178:

We also attempted using the average of mRFEI as a cutoff point to create a food swamp variable and found similar results in terms of the likelihood of residing in a food swamp/desert and relationship with diet quality.

  1. Table 3: Ordinal logistic regression: Does the essential Parallel Regression Assumption hold? If not, use a multinomial regression model instead or even better, run OLS on the metric diet quality score (see point 2).

Thank you for this excellent suggestion. To address your question, we ran the test of the parallel line for all three outcomes (i.e., diet, weight, health) including both full sample and race/ethnicity subgroups. We found that the assumption is plausible for the models with diet quality and health quality as outcome variables but not for the weight status. Therefore, we chose the ordinal logistic regression for models with diet quality and health quality as outcome variables, and multinomial regression models for weight status.

See lines 141-147 (below) for changes made.

To examine the relationship between the food environment and the three main outcome variables in both the full sample and three subsamples (i.e., non-Hispanic White, non-Hispanic Black, and Hispanic), we first checked to see if the ordinal logistic regression parallel line assumption held. The ordinal logistic regression model was then chosen to assess the predicted probabilities of perceived diet quality and health quality. However, since the assumption of parallel trend test was violated for perceived weight status, multinomial logistic regression models were chosen for this particular outcome variable.

  1. Analyses on weight status are missing completely. However, for further investigation into the external validity of mRFEI, regression of weight status on mRFEI would be worthwhile.

This is an excellent point. We ran the analysis on weight status but we didn’t find significant relationships in the full sample or any subsample between weight status and neighborhood environment (measured by mRFEI). Thus, we didn’t include the results table in our paper. However, we mentioned these results in the text.

 See lines 187-191 (copied below):

There were no statistically significant differences in the relationship between residing in a food swamp or food desert and diet quality in the Hispanic subsample or perceived health quality and weight status by food swamp/desert residential status in the total sample or any subsample. Therefore, we didn’t include those results in the result table. Therefore, we didn’t include the results table in our paper.

  1. 247-257: A further limitation of the study is that it is only able to identify correlations not causal effects, since the underlying data is cross-sectional.

Thank you for pointing this out. We have added a sentence to highlight this limitation. Further, the current study was only able to examine correlational relationships as the underlying dataset is cross-sectional. Future work should pursue focus on identifying causal effects as well as longitudinal designs and natural experiments to explore causality.

Lines 274-275 now read as follows:

Further, the current study was only able to examine correlational relationships as the underlying dataset is cross-sectional. Future work should pursue longitudinal designs and natural experiments to explore causality.

Reviewer 2 Report

This paper examines the diet quality of people living in food swamps versus food deserts, stratifying by race.

Introduction- What is the rationale for considering race an effect modifier? The fact that you used race as an effect modifier did not become apparent to me until later on in the paper. I think it should be laid out more clearly up-front.

The authors claim that the study is nationally representative. What makes it so? Were weights applied to the analysis to make the sample nationally representative?

Why did participants only receive 70 cents for participation?

Statistical approach

I don't agree with the use of binary logistic regression for the primary analysis. You really have a 3-category outcome, so a multinomial regression would be more appropriate. Otherwise when you use the binary logistic regression, you are combining 2 of your categories that don't make sense to be combined (i.e. food desert vs food swamp + neither and then food swamp vs food desert + neither).

Please state explicitly that analyses were stratified by race.

You need to describe the analysis with diet quality as the outcome.

Results:

Could you comment on the fact that only 5% had low diet quality? This seems low. How does this compare to other populations that may have used the same instrument? Did you consider assessing diet quality using a different cut-off?

Table 2 results- I don't understand how these results represent stratified models. If it's stratified by race/ethnicity, how can you compare between the races? For example, what does the OR of 0.70 for Non-Hispanic White represent? Does it mean that Non-Hispanic Whites had a 0.70 lower odds of living in a Food Swamp Area compared to other races? If so, then those results do not come from stratified models, since a stratified model would have only Non-Hispanic Whites in it. Perhaps the issue is an inconsistent use of the word "stratified" since in Table 3, it does look like these values come from stratified models.

When you're comparing ORs across different races, please list the ORs for both races in order to compare. For example, in line 171 it doesn't help the reader to only know the OR for non-Hispanic Blacks without also listing the OR for Black Americans.

In the methods you said that you ran interaction tests. Please list the P values for interaction as a final column in Table 3, or at the very least within the written results.

Discussion:

Line 234: Was weight status an outcome in the study? If so it needs to be described in the methods and the results.

Minor wording/typographical errors:

Line 67: "the" missing before the word following

Line 83: "d" missing in "agreed"

Line 144: "at" should be "had"

Line 188: choose either "varies" or "differ" but not both

Author Response

September 9, 2020

Dear Reviewer,

Thank you for allowing us the opportunity to submit a revised draft of our manuscript titled “Racial Differences in Perceived Food Swamp and Food Desert Exposure & Disparities in Self-Reported Dietary Habits” to the International Journal of Environmental Research and Public Health special issue devoted to Food Systems, Food Environment, Diet and Nutrition Related Diseases.

We appreciate the time that you and the reviewers have dedicated to reviewing the manuscript and providing edits. We have made revisions based on each of your suggestions in this revised version of the manuscript. Our responses are detailed below each of your initial comments.

Reviewer 2

  1. Introduction- What is the rationale for considering race an effect modifier? The fact that you used race as an effect modifier did not become apparent to me until later on in the paper. I think it should be laid out more clearly up-front.

Thank you for this important question. Our rationale stems from prior research. We have added the following sentence and citations to the introduction:

Lines 45-47 now read as follows:

Prior research also points to strong linkages between race and ethnicity, exposure to food swamps and other inequitable food environments, and dietary disparities.

  1. The authors claim that the study is nationally representative. What makes it so? Were weights applied to the analysis to make the sample nationally representative?

We are grateful to the reviewer for highlighting our error. We did not apply weights for the statistical analysis. This is a national sample of US adults. It is representative of the broader US adult population in terms of gender, educational attainment, income, and U.S. region but oversamples for Black Americans. We have revised the text to describe it as a “national sample of U.S. adults” in the abstract and in lines 79 and 197 of the manuscript.

  1. Why did participants only receive 70 cents for participation?

Thank you for this excellent question. MTurk is frequently used to administer surveys and experiments at a low cost. Mturkers were recruited via a job posting (comparable to a recruitment flyer) which described the survey (e.g., incentive amount, topic, completion time etc.). Participants were able to opt-in to taking the online survey by clicking on the survey job posting listing. We revised the manuscript to cite several articles using comparable incentives for Mturkers (Each participate received from 0.5-1 dollars).

Lines 84-86 now read as follows:

In line with other studies using MTurk paying each participant $0.50 to $1.00, 42–45 we pay $0.70 to each MTurk online participant to complete the survey task.

Statistical approach

  1. I don't agree with the use of binary logistic regression for the primary analysis. You really have a 3-category outcome, so a multinomial regression would be more appropriate. Otherwise when you use the binary logistic regression, you are combining 2 of your categories that don't make sense to be combined (i.e. food desert vs food swamp + neither and then food swamp vs food desert + neither).

We really appreciate this excellent point. We agree with the reviewer. To address, this point, we redid the analysis as a multinomial regression instead. The reference group of the new model is non-food swamp/desert. Multiple comparisons across different race/ethnicity groups were also conducted. The statistical approach section (lines 132-152) now reads as follows:

Multinomial logistic regression models were employed to assess the likelihood of living in a perceived food swamp area or food desert area (measured with mRFEI) with living in a non-food swamp/desert area as the reference group. To assess racial and ethnic differences relative to non-Hispanic Whites and each racial and ethnic minority group, separate multinomial regression models including the following reference groups: non-Hispanic White, non-Hispanic Black, non-Hispanic Asian, non-Hispanic Other, and Hispanic. The estimates from the multinomial regression tests were then used to compute racial and ethnic differences in the relative risk (i.e., probability) of living in a food swamp/desert area.

To examine the relationship between the food environment and the three main outcome variables in both the full sample and three subsamples (i.e., non-Hispanic White, non-Hispanic Black, and Hispanic), we first checked to see if the ordinal logistic regression parallel line assumption held. The ordinal logistic regression model was then chosen to assess the predicted probabilities of perceived diet quality and health quality. However, since the assumption of parallel trend test was violated for perceived weight status, multinomial logistic regression models were chosen for this particular outcome variable. Covariates included family income; race/ethnicity; education level; current family structure; geographic area; car ownership; gender; region of the country; age, together with perceived food swamp and food desert exposure in each model. For all three main outcome variables, analyses were also stratified by race and ethnicity. Further, for models including the full sample of respondents, we added the interaction effect between income and race. We ran all statistical analyses using SPSS version 24.50

We also employed multinomial logistic regression analyses to recalculate the relative risk ratio across different races (food swamp vs. non-food swamp/desert & food desert vs. non-food swamp/desert). See table 2 and lines 166-176 (copied below).

Refer to Table 2 for the likelihood that participants perceived living in a food swamp or food desert (measured using mRFEI), by race and ethnicity. The results of multinomial logistic regression models showed that compared to non-Hispanic Whites, non-Hispanic Blacks were 38% more likely to perceive living in a food swamp area (RR=1.38, p<.001); non-Hispanic Asians were less likely to perceive living in a food swamp than Asians (RR=0.70, p<.05). Compared to non-Hispanic Blacks, non-Hispanic Asians were less likely to perceive living in both food swamp (RR=0.53, p<.001) and food desert area (RR=0.13, p<.05); and Hispanics were less like to perceive living in a food swamp area (RR=0.8, p<.05). Compared to non-Hispanic Asians, non-Hispanic Others were more likely to perceive living in both food swamp (RR=1.59, p<.05) and food desert areas (RR=7.52, p<.05). Last, Hispanics were more likely to perceive living in a food swamp area than non-Hispanic Other (RR=1.50, p<.05). See Table 2.

  1. Please state explicitly that analyses were stratified by race.

Thank you for this suggestion. We have revised this manuscript to explicitly state that the analyses were stratified by race. Lines 141-151 now reads as follows:

To examine the relationship between the food environment and the three main outcome variables, we first checked to see if the ordinal logistic regression parallel line assumption held. The ordinal logistic regression model was then chosen to assess the predicted probabilities of perceived diet quality and health quality. However, since the assumption of parallel trend test was violated for perceived weight status, multinomial logistic regression models were chosen for this particular outcome variable. Covariates included family income; race/ethnicity; education level; current family structure; geographic area; car ownership; gender; region of the country; age, together with perceived food swamp and food desert exposure in each model. For all three main outcome variables, analyses were also stratified by race and ethnicity. Further, for models including the full sample of respondents, we added the interaction effect between income and race. We ran all statistical analyses using SPSS version 24.49

  1. You need to describe the analysis with diet quality as the outcome.

Thank you for this excellent suggestion. We reorganized the section describing the analysis with diet quality as the outcome.  Lines 141-152 now read as follows:

“…. we first checked to see if the ordinal logistic regression parallel line assumption held. The ordinal logistic regression model was then chosen to assess the predicted probabilities of perceived diet quality and health quality. Covariates included family income; race/ethnicity; education level; current family structure; geographic area; car ownership; gender; region of the country; age, together with perceived food swamp and food desert exposure in each model. For all three main outcome variables, analyses were also stratified by race and ethnicity. Further, for models including the full sample of respondents, we added the interaction effect between income and race. We ran all statistical analyses using SPSS version 24.49

Results:

  1. Could you comment on the fact that only 5% had low diet quality? This seems low. How does this compare to other populations that may have used the same instrument? Did you consider assessing diet quality using a different cut-off?

Thanks for pointing out this very important issue. We agree with the review that the way that we initially categorized the variable was not appropriate. We revised our approach to draw from the distribution of the diet quality composite score to create a diet quality variable with the following 3 different categories: low (31.0%); medium (36.0%); high (31.3%).

See lines 102-105 for the changes made:

Finally, we collapsed categories based on the range distribution of diet quality composite score to create a high (124 to 27, 31.0%), medium (71 to 11, 36.0%3), and a low category (-12 to 6, 31.3%0) for diet quality which we used in the analysis.  

  1. Table 2 results- I don't understand how these results represent stratified models. If it's stratified by race/ethnicity, how can you compare between the races? For example, what does the OR of 0.70 for Non-Hispanic White represent? Does it mean that Non-Hispanic Whites had a 0.70 lower odds of living in a Food Swamp Area compared to other races? If so, then those results do not come from stratified models, since a stratified model would have only Non-Hispanic Whites in it. Perhaps the issue is an inconsistent use of the word "stratified" since in Table 3, it does look like these values come from stratified models.

Thank you for pointing this out. We agree that our initial use of the word “stratified” was confusing. To address the reviewer’s point, we deleted the word “stratified” when describing the results of Table 2 and used multinomial regression analysis instead. The new models were not stratified by race/ethnicity.

To clarify these points, the section describing the Methods and the Results in Table 2 (lines 167-177) now read as follows:

Lines 134-139:

To assess racial and ethnic differences relative to non-Hispanic Whites and each racial and ethnic minority group, separate multinomial regression models including the following reference groups: non-Hispanic White, non-Hispanic Black, non-Hispanic Asian, non-Hispanic Other, and Hispanic. The estimates from the multinomial regression tests were then used to compute racial and ethnic differences in the relative risk (i.e., probability) of living in a food swamp/desert area.

Lines 167-177:

Refer to Table 2 for the likelihood that participants perceived living in a food swamp or food desert (measured using mRFEI), by race and ethnicity. The results of multinomial logistic regression models showed that compared to non-Hispanic Whites, non-Hispanic Blacks were 38% more likely to perceive living in a food swamp area (RR=1.38, p<.001); non-Hispanic Asians were less likely to perceive living in a food swamp than Asians (RR=0.70, p<.05). Compared to non-Hispanic Blacks, non-Hispanic Asians were less likely to perceive living in both food swamp (RR=0.53, p<.001) and food desert area (RR=0.13, p<.05); and Hispanics were less like to perceive living in a food swamp area (RR=0.8, p<.05). Compared to non-Hispanic Asians, non-Hispanic Others were more likely to perceive living in both food swamp (RR=1.59, p<.05) and food desert areas (RR=7.52, p<.05). Last, Hispanics were more likely to perceive living in a food swamp area than non-Hispanic Others (RR=1.50, p<.05).

  1. When you're comparing ORs across different races, please list the ORs for both races in order to compare. For example, in line 171 it doesn't help the reader to only know the OR for non-Hispanic Blacks without also listing the OR for Black Americans.

 Thank you for this important question. We can see how our original wording was unclear. We didn’t compare ORs across different races in the paper. Instead, we calculated relative risk ratios across different race/ethnicity groups. We have added additional details in the Methods and Results sections. Lines 132-139 and 167-177 now read as follows:

Lines 132-139

Multinomial logistic regression models were employed to assess the likelihood of living in a perceived food swamp area or food desert area (measured with mRFEI) with living in a non-food swamp/desert area as the reference group. To assess racial and ethnic differences relative to non-Hispanic Whites and each racial and ethnic minority group, separate multinomial regression models including the following reference groups: non-Hispanic White, non-Hispanic Black, non-Hispanic Asian, non-Hispanic Other, and Hispanic. The estimates from the multinomial regression tests were then used to compute racial and ethnic differences in the relative risk (i.e., probability) of living in a food swamp/desert area.

Lines 167-177

Refer to Table 2 for the likelihood that participants perceived living in a food swamp or food desert (measured using mRFEI), by race and ethnicity. The results of multinomial logistic regression models showed that compared to non-Hispanic Whites, non-Hispanic Blacks were 38% more likely to perceive living in a food swamp area (RR=1.38, p<.001); non-Hispanic Asians were less likely to perceive living in a food swamp than Asians (RR=0.70, p<.05). Compared to non-Hispanic Blacks, non-Hispanic Asians were less likely to perceive living in both food swamp (RR=0.53, p<.001) and food desert area (RR=0.13, p<.05); and Hispanics were less like to perceive living in a food swamp area (RR=0.8, p<.05). Compared to non-Hispanic Asians, non-Hispanic Others were more likely to perceive living in both food swamp (RR=1.59, p<.05) and food desert areas (RR=7.52, p<.05). Last, Hispanics were more likely to perceive living in a food swamp area than non-Hispanic Others (RR=1.50, p<.05).

  1. In the methods you said that you ran interaction tests. Please list the P values for interaction as a final column in Table 3, or at the very least within the written results.

We appreciate your suggestion. The results of the Race X Income interactions are included in the full model provided in Appendix Table A (copied below). As we didn’t find significant results for these interaction tests these results were not included in the abbreviated Table 3 (i.e., Low Income X Black, Low Income X Asian, Low Income X Hispanic, and Low Income X Other all had p values>.05).

Appendix Table A. Summary of logistic regression models predicting diet quality by residing in a food swamp or food desert (measured using mRFEI), stratified by race. **FULL MODEL**

                All

Non-Hispanic White (N=2912)

Non-Hispanic Black (N=954)

Hispanic (N=162)

Independent variables/covariates

OR

95%CI

OR

95% CI

OR

95% CI

OR

95% CI

lower

upper

lower

upper

lower

upper

lower

upper

Residing in Food swamp1

.75***

.66

.84

.75***

.64

.87

.66**

.51

.86

1.53

.79

2.96

Residing in Food desert

.74*

.58

.94

.75*

.56

.99

.81

.48

1.39

.25

.05

1.27

Lower-income (vs. higher income)

.86*

.74

.99

.85*

.73

.99

.90

.68

1.18

1.51

.78

2.93

Non-Hispanic Black 2

.66*

.53

.82

-

-

-

-

-

-

-

-

-

Non-Hispanic Asian

.83

.49

1.41

-

-

-

-

-

-

-

-

-

Non-Hispanic Other

1.51

.98

2.30

Hispanic

.86

.56

1.31

-

-

-

-

-

-

-

-

-

High school or less3

.45***

.39

.53

.45***

.37

.54

.58**

.38

.87

.46

.17

1.29

Associate degree and some college

.75***

.65

.85

.76**

.64

.89

.74*

.56

.97

.97

.48

1.98

Single without children4

.89

.60

1.32

.95

.56

1.61

.84

.43

1.65

.68

.08

6.02

Single with children

.81

.53

1.23

.84

.47

1.48

.77

.38

1.58

1.11

.09

14.09

Married with children

1.28

.82

1.98

1.3

.75

2.30

1.28

.53

3.14

2.45

.23

25.81

Life partner without children

1.11

.74

1.67

1.19

.70

2.02

.92

.45

1.87

.40

.04

4.25

Life partner with children

1.08

.67

1.74

1.15

.63

2.10

.81

.32

2.09

1.30

.08

21.15

Own a car or someone in my house own a car (vs. do not own a car)

1.16

.92

1.45

1.29

.93

1.79

.97

.68

1.37

3.42

.61

19.22

Male (vs. female)

.77***

.69

.87

.76**

.66

.88

.83

.63

1.09

.54

.28

1.06

Midwest5

.72**

.60

.87

.73**

.59

.90

.73

.47

1.14

.621

.20

1.96

Northeast

.91

.75

1.09

.93

.74

1.17

.71

.47

1.09

1.23

.46

3.34

Southeast

.72***

.60

.85

.73**

.59

.90

.58**

.39

.86

1.51

.64

3.55

Southwest

.80

.64

1.00

.88

.67

1.17

.52*

.29

.90

.66

.25

1.74

Urban6

.89

.75

1.05

.98

.80

1.21

.70

.46

1.06

.64

.24

1.70

Suburban

.95

.82

1.10

.91

.77

1.07

1.04

.68

1.56

.62

.23

1.67

Age

1.01***

1.01

1.01

1.01*

1.00

1.01

1.02***

1.01

1.03

1.02

.99

1.05

Low income* Black/

.19

.01

2.87

-

-

-

-

-

-

-

-

-

Low income * Asian

1.12

.84

1.48

-

-

-

-

-

-

-

-

-

Low income * Other

1.29

.56

2.97

-

-

-

-

-

-

-

-

-

Low income * Hispanic

.72

.40

1.29

-

-

-

-

-

-

-

-

-

1Reference group is non-food swamp/desert areas; 2Reference group is Non-Hispanic White; 3Reference group is Bachelor’s degree or higher; 4Reference group is married without children 5Reference group is West; 6Reference group is rural; 7Reference group is middle/high income * White; Outcome variable diet quality has three mutually exclusive categories: low, medium, high; Health quality has five mutually exclusive categories: poor, fair, good, very good, excellent; weight status has four mutually exclusive categories: slightly underweight, about right, slightly overweight, very overweight.; Abbreviations: OR, odds ratios; CI, confidence interval; *p<.05; **p<.01; ***p<.001.

Discussion:

  1. Line 234: Was weight status an outcome in the study? If so it needs to be described in the methods and the results.

Thanks for pointing this out. We did run the analysis on weight status and included it in our methods and results, but we didn’t find significant relationships between weight status and neighborhood food environment measured by mRFEI. Thus, we didn’t include the results table in our paper.

To clarify these points, the analysis and results sections have been revised as follows:

Lines 145-151:

However, since the assumption of parallel trend test was violated for perceived weight status, multinomial logistic regression models were chosen for this particular outcome variable. Covariates included family income; race/ethnicity; education level; current family structure; geographic area; car ownership; gender; region of the country; age, together with perceived food swamp and food desert exposure in each model. For all three main outcome variables, analyses were also stratified by race and ethnicity. Further, for models including the full sample of respondents, we added the interaction effect between income and race.

Lines 188-192:

There were no statistically significant differences in the relationship between residing in a food swamp or food desert and diet quality in the Hispanic subsample or perceived health quality and weight status by food swamp/desert residential status in the total sample or any subsample. Therefore, we didn’t include the results table in our paper.

Minor wording/typographical errors:

  1. Line 67: "the" missing before the word following

Thanks for your pointing this out. We have fixed this. Now on line 69.

  1. Line 83: "d" missing in "agreed"

Thanks for your comment. We have fixed this. Now on line 87.

  1. Line 144: "at" should be "had"

Thanks for your comment. We have fixed this. Now on line 161.

  1. Line 188: choose either "varies" or "differ" but not both

Thank you for pointing this out. We have fixed this. Now on line 203.

Round 2

Reviewer 2 Report

Thank you very much for addressing my comments. The only thing remaining is to carefully proofread the manuscript. For example, the following sentence is not a full sentence: To assess racial and ethnic differences relative to non-Hispanic Whites and each racial and ethnic minority group, separate multinomial regression models including the following reference groups: non-Hispanic White, non-Hispanic Black, non-Hispanic Asian, non-Hispanic Other, and Hispanic.

It should read "...separate multinomial regression models were run..."

Another sentence reads: Therefore, we didn’t include the results table 191 in our paper.

I would avoid the use of contractions for scientific writing.

Author Response

September 17, 2020

Dear Editor,

Thank you for allowing us the opportunity to submit a revised draft of our manuscript titled “Racial Differences in Perceived Food Swamp and Food Desert Exposure & Disparities in Self-Reported Dietary Habits” to the International Journal of Environmental Research and Public Health special issue devoted to Food Systems, Food Environment, Diet and Nutrition Related Diseases.

We appreciate the time that you and the reviewers have dedicated to reviewing the manuscript. We have made the requested revisions. Our specific edits are described below.

Reviewer 2

Thank you very much for addressing my comments. The only thing remaining is to carefully proofread the manuscript. For example, the following sentence is not a full sentence: To assess racial and ethnic differences relative to non-Hispanic Whites and each racial and ethnic minority group, separate multinomial regression models including the following reference groups: non-Hispanic White, non-Hispanic Black, non-Hispanic Asian, non-Hispanic Other, and Hispanic.

It should read "...separate multinomial regression models were run..."

Another sentence reads: Therefore, we didn’t include the results table 191 in our paper.

I would avoid the use of contractions for scientific writing.

-Thank you for this excellent suggestion. We have carefully proofread the manuscript and uploaded the revised versions. Detailed sections are line numbers are listed below.

  • Abstract: Both food swamps and food deserts have been associated with racial, ethnic, and socioeconomic disparities in obesity rates. Little is known about how the distribution of food deserts and food swamps relate to disparities in self-reported dietary habits, and health status, particularly for historically marginalized groups. In a national U.S. sample of 4,305 online survey participants (age 18+), multinomial logistic regression analyses were used to assess the likelihood of living in a food swamp or food desert area by race and ethnicity. Predicted probabilities of self-reported dietary habits, health status, and weight status were calculated using the fitted values from ordinal or multinomial logistic regression models adjusted for relevant covariates. Results showed that non-Hispanic, Black participants (N=954) were most likely to report living in a food swamp. In the full and white subsamples (N=2912), the perception of residing in a food swamp/desert was associated with less-healthful self-reported dietary habits overall. For non-Hispanic Blacks, regression results also showed that residents of perceived food swamp areas (OR=.66, p<.01, 95% CI [.51, .86]) had a lower diet quality than those in a non-food swamp and food desert area. Black communities in particular may be at risk for environment-linked diet-related health inequities. These findings suggest that an individual’s perceptions of food swamp and food desert exposure may be related to diet habits among adults.
  • Lines 44-45: Previous studies suggest that food swamps are strong predictors of disparities in obesity prevalence among adults.10
  • Lines 56-63: Recent studies utilizing self-reported food environment measures found significant associations between healthy food access and diet quality.26–34 However, little is known about how perceived food environment relates to self-reported dietary intake and related health, particularly for racial and ethnic minority groups in the US. This study sought to understand how perceptions of food store availability and accessibility are related to reported diet, weight status, and general health. In particular, the study explores whether living in a perceived food desert and food swamp was more likely to be reported by lower-income or racial and ethnic minority individuals and if such perceptions are related to lower-quality diets, higher weight, and worse reported health.

  • Lines 65-72: The current study is based on an online research sample. Study participants came from Amazon Mechanical Turk (MTurk), a crowdsourcing website which is widely used to obtain high-quality data rapidly and inexpensively35–38 This study utilizes data from 30 survey questions which was distributed on Qualtrics assessing 1) Food store access 2) Dietary habits 3) Perceived Weight status 4) Perceived Health Status, and 5) Demographics. The survey also included the following red herring question to identify whether participants fully read and engaged in the survey or not, “how closely do you read survey questions? To show that you're paying attention, please click the "Not closely at all" option as your answer.” The survey was in English only.

  • Lines 79-88: The original national sample of U.S. adults consisted of 6357 participants. Drawing from previous research on enhancing data quality in online survey research by screening for inattentive respondents and “speedy” completion times, participants were excluded if they wrongly answered “red herring” questions40-42(609 participants excluded). Relatedly, participants were also excluded if their response time was less than 6.5 minutes (1443 participants excluded based on the quartile), indicating that they did not answer the questions carefully.41,42 The final sample consisted of 4305 participants. In line with other studies paying “MTurkers” $0.50 to $1.00 to complete a survey task, 43–46 we paid $0.70 to each MTurk online survey participant. The Institutional Review Boards of Duke University approved all study procedures and materials. Respondents signed an online consent form to indicate that they agreed to participate in the study.

  • Lines 102-107: Finally, we collapsed categories based on the distribution of diet quality composite score to create a high (12 to 27, 31.0%), medium (7 to 11, 36.0%), and a low category (-12 to 6, 31.3%) for diet quality which we used in the analysis. Self-reported weight status was measured in a question asking participants how they described their weight choosing from the following responses: underweight, about right, slightly overweight, and very overweight. To assess self-reported health quality, we used a question asking participants to rate their health quality as one of 5 different levels including poor, fair, good, very good, and excellent.

  • Lines 131-136: Multinomial logistic regression models were employed to assess the likelihood of living in a perceived food swamp area or food desert area (measured with mRFEI). The reference group included individuals living in a non-food swamp/desert area. To assess racial and ethnic differences relative to non-Hispanic Whites and between each racial and ethnic minority group, separate multinomial regression models were run including the following reference groups: non-Hispanic White, non-Hispanic Black, non-Hispanic Asian, non-Hispanic Other, and Hispanic.

  • Lines 146-150: For all regression models, covariates included: family income; race/ethnicity; education level; current family structure; geographic area; car ownership; gender; region of the country; age, with perceived food swamp and food desert exposure in each model. For all three main outcome variables, analyses were also stratified by race and ethnicity. Further, for models including the full sample of respondents, we included race and income interaction terms.

  • Lines 153-159: The final sample consisted of 4305 participants with 38.0% male, 67.6% non-Hispanic White, 22.2% non-Hispanic Black, 2.0% non-Hispanic Asian, 3.8% Hispanic, 33.6% single, and 30.8% low income. Participants were on average 41.3 years old and about half (48.8%) of them lived in a suburban area. Based on mRFEI measures, 40.7% of study participants reported a retail environment consistent with living in a food swamp, and 6.5% reported a retail environment consistent with living in a food desert. More than half of study participants (58.4%) owned a car or someone in the house owned a car. 41.3% of respondents had a bachelor’s degree or more.

  • Lines 187-190: There were no statistically significant differences in the relationship between residing in a food swamp or food desert and diet quality in the Hispanic subsample or perceived health quality and weight status by food swamp/desert residential status in the total sample or any subsample. Therefore, we did not include the results in our paper.

  • Lines 229-235: This finding suggests that an individual’s perceptions of food swamp and food desert exposure may be an important correlate of diet quality. However, there is limited evidence supporting the elimination of food deserts as a strategy to address poor diet quality.61 A recent study found that opening a supermarket in a food desert did not result in people buying healthier food.62 Recent studies of neighborhood food environment are shifting their focus to the culpability of food swamps9,51 which may be a stronger predictor of inequities in diet and obesity than food deserts.10,11
  • Lines 247-260: There were no statistically significant differences in self-reported health quality and weight status by food swamp residential status in our study. Though there is limited research assessing the relationship between the food environment and perceived health, evidence has suggested diet quality has a strong impact on health outcomes.1,65–68 Previous studies assessing the association between the neighborhood environment and weight status were inconsistent.6,17,68–82 Similar to our study, several studies have not found a statistically significant relationship between neighborhood food environment and obesity rates.72–75 However, other studies have shown that neighborhood food environment factors, such as food desert and food swamp status, were significantly associated with individuals’ measured weight status.10,11 For example, Chen et al used 7 MedProfiler questions about dietary behaviors (e.g., frequency of consuming high-fiber, high-protein, low-carbohydrate, low-fat, low-salt, and low-sugar diets) and found, even after we controlling for home food environment factors, food desert status was associated with obesity.68 The different findings may be due to error inherent in self-reported measures of weight status, diet quality and researchers’ varied ways in measuring the built, food environment.68
  • Lines 267-270: Also, the relatively small sample size of Hispanic participants and the absence of acculturation and country of origin limits our understanding of Hispanic ethnicity as a moderator of the relationship between food swamps and diet and may explain why we did not find statistically significant results in this subsample.